# Thrombospondin-Related Anonymous Protein (TRAP) Family Expression by *Babesia bovis* Life Stages within the Mammalian Host and Tick Vector

**DOI:** 10.3390/microorganisms10112173

**Published:** 2022-11-02

**Authors:** Hayley E. Masterson, Naomi S. Taus, Wendell C. Johnson, Lowell Kappmeyer, Janaina Capelli-Peixoto, Hala E. Hussein, Michelle R. Mousel, Diego J. Hernandez-Silva, Jacob M. Laughery, Juan Mosqueda, Massaro W. Ueti

**Affiliations:** 1Program in Vector-Borne Diseases, Department of Veterinary Microbiology and Pathology, Washington State University, Pullman, WA 99164, USA; 2Animal Diseases Research Unit, Agricultural Research Service, US Department of Agriculture, Pullman, WA 99164, USA; 3Immunology and Vaccines Laboratory, Facultad de Ciencias Naturales, Universidad Autónoma de Queretaro, Queretaro 76230, Mexico

**Keywords:** *Babesia*, infection, TRAP, ticks, transmission, cattle

## Abstract

The tick-transmitted disease bovine babesiosis causes significant economic losses in many countries around the world. Current control methods include modified live-attenuated vaccines that have limited efficacy. Recombinant proteins could provide effective, safe, and low-cost alternative vaccines. We compared the expression of the *Babesia bovis* thrombospondin-related anonymous protein (TRAP) family from parasites in bovine blood, in vitro induced sexual stages, and kinetes from tick hemolymph. Quantitative PCR showed that in blood and sexual stages, TRAP3 was highly transcribed as compared to the other TRAPs. In contrast, the TRAP1 gene was highly transcribed in kinetes as compared to the other TRAPs. Fixed immunofluorescence assays showed that TRAP2, 3, and 4 proteins were expressed by both blood and sexual stages. Conversely, TRAP1 protein, undetected on blood and induced sexual stages, was the only family member expressed by kinetes. Live IFA revealed that TRAP2, 3, and 4 proteins were expressed on the surface of both *B. bovis* blood and sexual stages. Modeling of *B. bovis* TRAP1 and TRAP4 tertiary structure demonstrated both proteins folded the metal-ion-dependent adhesion site (MIDAS) domain structure of *Plasmodium* TRAP. In conclusion, TRAP proteins may serve as potential vaccine targets to prevent infection of bovine and ticks with *B. bovis* essential for controlling the spread of bovine babesiosis.

## 1. Introduction

Bovine babesiosis is a disease caused by intraerythrocytic parasites, including *Babesia bovis*, *B. bigemina*, and *B. divergens* [1,2,3]. This disease is common in tropical and subtropical climates, causing significant economic losses in many countries [4,5]. Infected bovines show signs of fever, anemia, and, in severe cases, death [3]. Climate change poses a significant threat to the control of bovine babesiosis as it allows the spread of the tick vector into non-endemic areas and evasion of current control strategies [6]. Therefore, there is a need for vaccines that effectively control this disease. Current methods for the control of bovine babesiosis include drugs, acaricide treatment, and live-attenuated vaccines [7,8,9]. Each method comes with its own challenges. Drugs are expensive, unable to prevent severe disease, and may leave chemical residues in milk and meat [3]. The utilization of acaricide treatments can lead to resistant tick populations and contamination of the environment [10]. Live vaccines are expensive, difficult to make, and may have a limited shelf life [11]. Live vaccines present further challenges such as the potential for contamination with other pathogenic organisms and the loss of immunogenicity [12]. It is imperative to develop a subunit vaccine that is less expensive, safer, with a longer shelf life, and more protective than the current live vaccines [13].

The life cycle of *Babesia* is complex and consists of important stages for parasite development within the mammalian host and tick vectors. Within the tick midgut lumen, sexual stages of the parasite are formed. *Babesia bovis* sexual stages express HAP2 protein, which plays an important role in fertilization and formation of the zygote [14]. A previous study demonstrated that HAP2 was expressed on the surface of *Plasmodium* microgametes and was important for the fusion of parasite sexual stages prior to infecting insect gut epithelial cells [15]. Zygotes infect tick midgut epithelial cells and transform into kinetes [16]. The kinetes mature within midgut cells before being released into tick hemolymph, gaining access to, and invading the ovaries. The parasites are transovarially transferred to the next generation of ticks [17]. To date, there are only a few reports regarding protein expression by different stages of *Babesia* parasites that cause bovine babesiosis [16,18]. These studies identified differentially expressed proteins that may allow the parasite to infect tick ovaries and invade eggs. Within tick larvae, the parasite forms a dense spherical body inside salivary gland cells [17]. When infected larvae feed on a bovine, the spherical bodies divide into sporozoites [19]. Sporozoites are inoculated via tick saliva into the mammalian host and directly infect erythrocytes, transforming into trophozoites and merozoites that lyse the erythrocyte, causing hemolytic anemia, and infecting other erythrocytes resulting in the persistent infection of the mammalian host [17,19]. Understanding protein expression by *B. bovis* blood and tick stages is critically important for the development of new control strategies.

A previous study suggested that the *B. bovis* thrombospondin-related anonymous protein (TRAP) family were potential antigen candidates for vaccine development [18]. These proteins are conserved among all Apicomplexan parasites [20,21]. During gliding movement, these proteins are involved in the moving junction, a structure composed of a few rhoptry proteins that form complexes with host surface proteins and parasite surface-exposed integral membrane microneme proteins [22]. The purpose of the vWFA and TSP-1 domains is to form cell-matrix interactions that assist in erythrocyte invasion [20,23,24]. Proteolytic cleavage of TRAP facilitates junction movement, releasing the parasite into the parasitophorous vacuole [22,23]. This vacuole is an invagination of the erythrocytic membrane that engulfs the parasite, allowing it entry to the cell [3]. Once the parasite is inside the cell, the parasitophorous vacuole degrades. Further investigation into the function and expression of TRAP genes is necessary to determine if these proteins are appropriate candidates for vaccine development. In this study, we evaluated the expression of the TRAP family, TRAP1 (BBOV_II002650), TRAP2 (BBOV_II002890), TRAP3 (BBOV_II002630) and TRAP4 (BBOV_II002870), by *B. bovis* parasite stages during its development within mammalian and tick hosts. We quantified transcript levels expressed by *B. bovis* blood stages, in vitro induced sexual stages, and kinetes derived from tick hemolymph. We determined protein expression and surface location using fixed and live immunofluorescence assays (IFAs), respectively. We also modeled the tertiary structure of TRAP1 and TRAP4 to characterize structural and functional attributes. Herein, we present the repertoire of TRAP expression by distinct life stages of *B. bovis* and discuss the importance of TRAP proteins for the parasite’s life cycle and their potential for vaccine development to block parasite transmission by the tick vector and prevent infection of the mammalian host.

## 2. Materials and Methods

### 2.1. Cattle, Pathogen, and Tick Vector

Three four-month-old splenectomized calves *Babesia*-free as determined as previously described [18] were used for the experiment. A *B. bovis* Texas strain, S74-T3Bo, and *Rhipicephalus microplus* ticks, La Minita strain, were also used. Briefly, *R. microplus* larvae were fed under a cloth patch on each calf. When nymphs started to molt to the adult stage, ~10^7^ *B. bovis*-infected erythrocytes were intravenously inoculated into the calves to synchronize tick feeding with the peak parasitemia [17,18]. Clinical signs of babesiosis were monitored daily, including fever and anemia. Giemsa-stained blood smears were used to determine the presence of parasites in calf blood.

The experiment conducted in this study was in accordance with the institutional guidelines based on the U.S. National Institutes of Health Guide for the Care and Use of Laboratory Animals and the Guide for the Care and Use of Agricultural Animals in Research and Training. This study was approved by the University of Idaho’s Institutional Animal Care and Use Protocol Committee, Moscow, Idaho, (IACUC #2018-16).

### 2.2. Babesia bovis Blood Stages

Blood from *B. bovis*-infected calves during acute parasitemia was collected into flasks containing glass beads and shaken. *Babesia bovis*-infected erythrocytes were grown in culture medium as previously described [18]. The cultures were incubated at 37 °C with 5% CO_2_ to allow optimum growth conditions. Infected *B. bovis* blood smears were generated to detect the expression of TRAP proteins by fixed IFA. Exoerythrocytic *B. bovis* merozoites were collected by differential centrifugation. Erythrocytes were pelleted at 400× *g* for 10 min. The supernatant was collected and centrifuged as before. Free merozoites remaining in the supernatant were pelleted from the final differential supernatant at 3000× *g* for 10 min. Exoerythrocytic parasites were used to determine surface-exposed proteins by a cell surface membrane staining assay of intact parasites. Moreover, *B. bovis*-infected cultures were centrifugated and the supernatant removed. TRIzol (Thermo Fisher Scientific, Waltham, MA, USA) was added to the infected erythrocyte pellet and stored at −80 °C.

### 2.3. Babesia bovis Induced Sexual Stages

Induction of *B. bovis* sexual stages was performed as previously described [14,25]. Microaerophilous stationary phase *B. bovis* cultures [26] at a parasitemia of 10% were suspended in induction medium for sexual stage development and incubated at 26 °C in air for up to 20 h or at 37 °C for 20 h as previously reported [14]. Induced sexual stages were collected to detect the expression of TRAP proteins by fixed and live IFAs. Additionally, induced sexual stages were collected, centrifuged, the pellet suspended in TRIzol, and stored at −80 °C for assessing TRAP transcription.

### 2.4. Babesia bovis Kinete Stage Isolation

Isolation of kinetes from tick hemolymph was performed as previously described [16]. Engorged female ticks were collected and incubated at 26 °C and 92% relative humidity to allow kinete accumulation in hemolymph as previously described [27]. Collected kinetes were stored in TRIzol at −80 °C. Kinetes were also collected in Hank’s Balanced Salt Solution to generate smears to detect the expression of TRAP proteins by fixed IFA.

### 2.5. RNA Isolation

Total RNA was extracted from TRIzol samples and treated with Turbo DNA-Free (Thermo Fisher Scientific) per the manufacture’s protocols. cDNA was synthesized from 100 ng of each total RNA using SuperScript III Reverse Transcriptase (Thermo Fisher Scientific). The number of individual TRAP transcripts expressed was determined using *Babesia* stage-specific cDNA.

### 2.6. Quantitative PCR

Triplicate PCR reactions were performed as previously described [18]. Quantitative PCR primers are presented in Table 1. The gene-family expression profile of TRAP was calculated by dividing the mean of each TRAP gene per mean of total TRAP transcripts and multiplied by 100. The expression profile compares gene expression by a single *Babesia* stage within a gene family without the need to normalize parasite numbers [28], thereby allowing assessment of differential expression by individual gene-family members between stages.

### 2.7. Fixed Immunofluorescence Assay

To detect stage-specific TRAP protein expression, smears of each stage were made on positively charged slides and stored at −80 °C as previously described [29]. Polyclonal antibodies were produced by immunizing rabbits with TRAP peptides (Table 2) predicted as surface-exposed moieties (GenScript, Piscataway, NJ, USA). Slides were stained for IFA as previously described [16]. A secondary conjugate of goat-anti-rabbit IgG Alexa Fluor 555 (Thermo Fisher Scientific) was used to detect specific antibody reactivity. ProLong Gold antifade reagent with 4,6-diamidino-2-phenylindole dihydrochloride (DAPI) (Thermo Fisher Scientific) was used to stain nuclei. A Leica microscope (Buffalo Grove, IL, USA) was used to examine antibody reactivity. Since HAP2 is expressed by *B. bovis* sexual stages but not blood stages, we used rabbit anti-HAP2 antibody as controls [14] to distinguish *B. bovis* sexual stages from blood stages.

### 2.8. Live Immunofluorescence Assay

To examine if TRAP proteins were expressed on the surface of live and intact cells, exoerythrocytic *B. bovis* merozoites from cultures and induced sexual stages [14,30] were washed in 10% BSA–PBS and incubated with 4 µg/mL of individual anti-TRAP primary antibodies for 30 min. The cells were washed in 10% BSA–PBS two times at 2000× *g* for 2 min to pellet the parasites. Washed cells were incubated with secondary conjugate antibody as above. The cells were washed two times at 2000× *g* for 2 min and incubated with 10 µg/mL 5(6)-Carboxyfluorescein Diacetate (5(6)-cFDA) (Sigma-Aldrich, St. Louis, MO, USA) and Hoechst 33342, trihydrochloride, trihydrate (Invitrogen, Waltham, MA, USA) in PBS. Washed samples were independently visualized as above. Rabbit anti-HAP2 antibody [14] was used as a control to detect surface exposed protein and to distinguish *B. bovis* sexual stages from blood stages.

### 2.9. Bioinformatic Analysis of TRAP Family Members

*Babesia bovis* TRAP family member polypeptide sequences were submitted for tertiary structure prediction to the Distance-guided Iterative Threading ASSEmbly Refinement (D-I-TASSER) server (https://zhanggroup.org//D-I-TASSER/, accessed on 27 August 2022). The D-I-TASSER relies on multiple deep neural network predictors that generate inter-amino acid residue interactions of contact maps, distance maps and hydrogen bond networks. Five models of tertiary structure prediction for the polypeptide were returned with a ranking of models based on a comparison to randomized D-I-TASSER predictions, and confidence in the models inferred by an eTM-score that was based on the strength of agreement of multiple D-I-TASSER simulations and contact map satisfaction rate. Threading technology was also paired with the distance-guided tertiary prediction in D-I-TASSER using 10 distinct threading technologies to match to structural templates in the Protein Data Bank (PDB: https://www.rcsb.org, accessed on 27 August 2022). A normalized Z-score was associated with the PDB threading hit generated by each threading program, with a Z-score > 1 indicating meaningful agreement in structure between the prediction model and PDB known structure. Model prediction results included a pdb file that contained information to generate a molecular model with contact distances and secondary structures that build the tertiary structure. The pdb file was opened using Chimera (http://www.cgl.ucsf.edu/chimera/, accessed on 27 August 2022) to view tertiary structures, and the Chimera MatchMaker feature used to overlay pdb files from TRAP predictions with those of PDB solved structures identified by threading.

### 2.10. Statistical Analysis

For within *B. bovis* development stage analysis, a full mixed linear model was evaluated including fixed effects of gene, plate, and technical replicate with random effect of the individual sample (tick or calf) in SAS 9.4 (SAS Inst. Inc., Cary, NC, USA). Plate and technical replicates were not significant and therefore removed from the final model. Pair-wise differences between genes within each of the three life stages was determined with a Tukey–Kramer adjustment. Gene-family profiling [28] was conducted to compare gene expression across tick stages. Differences across tick stages were determined with a full mixed linear module with fixed effects of life stage and gene and random effect of the individual sample in SAS 9.4. Pair-wise differences between genes across each of the life stages were determined with a Tukey–Kramer adjustment.

## 3. Results

### 3.1. Quantitative PCR

Within an individual *B. bovis* stage, expression of TRAP genes was significantly different (*p* < 0.05) (Table 3). In blood stages, there was no pair-wise statistical difference between gene expression of TRAP1 and TRAP4. However, TRAP2 and TRAP3 were statistically different for all pair-wise comparisons. TRAP gene expression by in vitro sexual stages showed there were no differences between TRAP1 and TRAP2; TRAP4 was different from TRAP2 but not different from TRAP1. There was no significant difference between TRAP3 and TRAP4 expression; however, expression of the TRAP3 gene was statistically different for all other pair-wise comparisons. In kinetes, there was no statistical difference between TRAP2, TRAP3, and TRAP4. In contrast, expression of TRAP1 was statistically different for all pair-wise comparisons.

In this study, we used gene-family expression profiles to calculate differences in the expression of TRAP family genes between *B. bovis* stages (Figure 1). The expression profile demonstrated that TRAP1 was significantly greater expressed by kinetes as compared to blood or sexual stages (*p* < 0.05). The expression of TRAP2 was significantly greater by *B. bovis* blood stages as compared to sexual stages or kinetes (*p* < 0.05). The expression of TRAP3 was significantly greater by *B. bovis* blood stages as compared to kinetes (*p* < 0.05) but not sexual stages (*p* = 0.06). The data suggested that TRAP1 may play an important function for *B. bovis* kinetes, whereas TRAP2 and 3 appear to be more important for *B. bovis* blood and sexual stages.

### 3.2. Expression of TRAP Proteins by B. bovis Blood Stages, Sexual Stages, and Kinetes

To test TRAP protein expression by *B. bovis* blood stages, sexual stages, or kinetes, antibodies against TRAP peptides were generated in rabbits and used in fixed or live IFA. Using fixed IFA, we demonstrated anti-TRAP2, 3, and 4 antibodies reacted with both *B. bovis* blood stages (Figure 2A) and in vitro induced sexual stages (Figure 3A). Matching pre-immune rabbit sera showed no antibody reactivity with *B. bovis* blood stages (Figure 2B) or in vitro induced sexual stages (Figure 3B).

The results indicated that TRAP2, 3, and 4 were expressed by *B. bovis* blood and sexual stages. Anti-HAP2 antibody, as expected, reacted only with *B. bovis* sexual stages (Figure 3A). Anti-TRAP1 antibody reactivity was undetectable using either *B. bovis* blood stages (Figure 2A) or in vitro induced sexual stages (Figure 3A). Matching pre-immune rabbit sera showed no antibody reactivity with *B. bovis* blood stages (Figure 2B) or in vitro induced sexual stages (Figure 3B).

These results suggested that TRAP1 was not expressed by *B. bovis* blood or sexual stages or was expressed at levels below the limit of detection of our assay. Using kinetes isolated from infected *R. microplus* females, anti-TRAP1 antibody reacted with fixed kinetes while anti-TRAP2, 3, 4, and HAP2 antibodies failed to react (Figure 4A). Matching pre-immune rabbit sera showed no antibody reactivity with *B. bovis* kinetes (Figure 4B).

Using live IFA, we demonstrated that TRAP2, 3, and 4 were surface-exposed proteins on both blood (Figure 5A) and in vitro induced sexual stages (Figure 6A). Matching pre-immune rabbit sera showed no antibody reactivity with *B. bovis* blood (Figure 5B) or in vitro induced sexual stages (Figure 6B). We were unable to determine if TRAP proteins were surface exposed due to the lack of methods to isolate intact live kinetes.

### 3.3. Bioinformatic Analysis of TRAP Family Members

Bioinformatic analysis demonstrated similar features and protein architecture with closely related hemoprotozoan pathogens of human beings that cause malaria. Sequence length restrictions of the D-I-TASSER server limited polypeptide length to 750 amino acid residues. Therefore, full length predictions were possible only for TRAP1 and TRAP4, but not for TRAP2 or TRAP3. For both *B. bovis* TRAP1 and TRAP4, threading hits with Z-scores greater than one generated by D-I-TASSER found several solved *Plasmodium* species TRAP structures in PDB, with the highest score of 3.99 to PDB 4hqlA, magnesium-loaded *Plasmodium vivax* TRAP protein [23]. The resulting pdb files from *B. bovis* TRAP1 (Figure 7A and Appendix A) and TRAP4 (Figure 7B) were overlayed with the pdb for 4hqlA to find visually appreciable overlap of the *Plasmodium* MIDAS (metal-ion dependent adhesion site) Mg^2+^ binding domain. Each helical and sheet structure in the *Plasmodium* domain was represented by an overlapping structure in the *Babesia* TRAP molecules.

A movie (Appendix A) showing the rotation of the overlapping molecular models reveals more completely the extent of the MIDAS domain structure similarity, while species-specific structural differences are also evident. The two *Plasmodium* TRAP molecules with solved structures in the PDB are AAA29775 (*P. falciparum*) and AAC97484 (*P. vivax*). The two *Plasmodium* proteins shared 43% identity, while *B. bovis* TRAP1 had 23% identity to either of the *Plasmodium* molecules. Cysteines forming disulfide bonds critical to the tertiary structure of the MIDAS binding domain in the *Plasmodium* TRAP molecules [23] were likewise conserved in *B. bovis* TRAP1 (Table 4). Further, key residues that bind the Mg^2+^ ion in the *Plasmodium* TRAP molecules were conserved in *B. bovis* TRAP1. *Babesia bovis* TRAP4 displayed reduced conservation but consistently formed the MIDAS binding domain and Mg^2+^ binding residues (Table 3).

*Babesia bovis* TRAP2 and TRAP3 truncated polypeptides were likewise submitted for D-I-TASSER analysis. However, neither TRAP2 or TRAP3 truncated versions generated significant Z-scores and did not return hits to any solved PDB structures related to TRAP, including 4hqlA. While key cysteines that form disulfide bonds of the MIDAS domain are largely conserved in *B. bovis* TRAP2 and TRAP3, the Mg^2+^ binding residues were not conserved (Table 3).

## 4. Discussion

Understanding protein expression by *B. bovis* life stages within bovine and ticks is a critically important step for developing strategies to protect bovine against babesiosis. Previous studies indicated that *B. bovis* TRAP proteins are potential candidates for vaccine development [20,21,24]. The structure of TRAP proteins is conserved among Apicomplexan parasites [21,22]. TRAP is defined as a protein that contains a vWFA region and TSP-1 domains [21]. The TRAP ectodomain, VWA domain, MIDAS ligand-binding site, and TSP-1 domain all play a role in *Plasmodium* motility [23]. In malaria, TRAP proteins are expressed by multiple stages of the parasite during its life cycle within vertebrate and arthropod hosts, including merozoites (MTRAP), ookinetes (CTRAP), and the salivary gland sporozoites (TRAP) [31]. *Plasmodium* TRAP proteins were used as vaccines against malaria infection of the vertebrate host or to block transmission by the biological vector [32,33]. Previous studies have also demonstrated that *Plasmodium* TRAP family proteins are responsible for parasite motility and invasion of host cells. In *Plasmodium* parasites, TRAP mediates gliding motility by attaching and detaching to different substrates along the surface of the parasites [23]. However, it is unknown if *B. bovis* TRAP proteins play important roles in parasite gliding movement or invasion of vertebrate and invertebrate host cells.

Our results demonstrated significant increased transcription levels of TRAP2 and 3 by *B. bovis* blood stages as compared to kinetes. However, TRAP3 in blood stages was similar to sexual stages. In contrast, TRAP1 was highly expressed by kinetes as compared to the other TRAP family members and *B. bovis* stages. It is clear from this and other studies that control of transcription is under the influence of various regulatory systems [18,34]. However, the mechanisms that control transcription in different parasite stages are not completely understood in protozoan parasites [34]. In this study using fixed IFA, we demonstrated that TRAP2, 3, and 4 proteins were expressed by *B. bovis* blood and sexual stages, whereas TRAP1 protein was only expressed by *B. bovis* kinetes. Like other Apicomplexa parasites, *B. bovis* TRAP proteins contain similar features, including a vWFA region. Due to this similarity, we propose that *B. bovis* TRAP proteins have a parallel function that facilitates parasite infection of bovine and tick vectors. A previous study demonstrated that all four TRAP proteins had signal peptides and transmembrane domains suggesting that they are predicted to be proteins exposed on the parasite’s surface [18]. Our live IFA results, corroborated with bioinformatic predictions, demonstrated that TRAP2, 3, and 4 were surface-exposed proteins on both *B. bovis* blood and sexual stages. Unfortunately, methods to isolate intact live kinetes are not available. A previous study using differential gel electrophoresis of surface-labeled proteins demonstrated that kinetes did not survive isolation intact and both cell membrane exterior surface and internal proteins were stained [16]. Therefore, we were unable to demonstrate the presence of surface-exposed proteins by kinetes. These collective data support the possibility that TRAP2, 3, and 4 proteins could be appropriate candidates for vaccine development to prevent infection of erythrocytes and to disrupt the formation of infectious *B. bovis* forms that infect tick midgut. A previous study showed that *B. bovis* TRAP2 was associated with the microneme at the apical end of the merozoite stage [24]. Zhan et al. found that *B. orientalis* TRAP2 was localized on the apical end of the parasite [35]. *Babesia bovis* TRAP2 shares 97% identity with *B. orientalis* TRAP2, indicating high conservation and a high likelihood of a similar function [35]. Furthermore, Terkawi et al. found that an antibody to rBbTRAP2 partially inhibited *B. bovis* growth in vitro by interfering with erythrocyte invasion [24]. In comparison to MSA-2c, RAP-1CT, and SBP-1, *B. bovis* TRAP2 antibodies had the greatest level of parasite growth inhibition [24]. Limited information is available for *B. bovis* TRAP3 and 4. However, these proteins may have a redundant function to infect bovine erythrocytes or be important for other stages of parasite development. TRAP2, 3, and 4 proteins were also detected in in vitro induced *B. bovis* sexual stages. *Babesia bovis* gametes are formed in the tick midgut lumen and fuse to form a zygote. We hypothesize that *B. bovis* TRAP proteins assist gamete motility through the midgut milieu to find their partners.

In this study, we demonstrated that, similar to *Plasmodium*, TRAP1 protein was made only by kinetes in the arthropod, suggesting that this protein may be involved in kinete motility and invasion of tick ovaries. However, previous studies have demonstrated that TRAP1 was expressed by *B. bovis* merozoites [20,36] and by *B. bigemina* blood stages [37]. The discrepancy between studies could be explained by *Babesia* species or strain diversity. In the current study, we used the *B. bovis* S74-T3Bo strain, while previous studies of *B. bovis* TRAP1 used either an Israel strain, clonal line C61411, or an alternative Texas strain [20,36]. This assumption is also supported by *Plasmodium* strain diversity that negatively impacted the detection of infection in human beings [38]. In addition, a serological survey using field collected samples from geographically distinct areas showed that TRAP1 induced inconsistent levels of antibodies, suggesting that the protein was either a poor immunogen or not expressed by *B. bovis* isolates [36]. Nonetheless, *B. bovis* TRAP1 could be used as an antigen for vaccine development to disrupt parasite transmission by tick vectors and prevent mammalian host infection.

In *Plasmodium* species, TRAP allows the parasite to invade mosquito tissues and mammalian host cells. Herein, we demonstrated that *B. bovis* TRAP1 was highly expressed by *Babesia* kinete parasites. This result is similar to those obtained in a previous study [39]. Even though *B. bovis* TRAP1 shares only 23% amino acid identity with the *Plasmodium* sp. TRAP protein, *B. bovis* TRAP1 appears to fold a similar MIDAS domain suggesting mechanistic conservation. Furthermore, the *B. bovis* TRAP4 gene was upregulated, and the protein made by both *B. bovis* blood and sexual stages. The presence of a MIDAS domain suggests that TRAP4 may be important for gliding and invasion of vertebrate and invertebrate host cells by *Babesia* blood and sexual stages. The location of *B. bovis* TRAP1 and TRAP4 predicted that rhomboid protease cleavage sites, AGGALLG and VGIICLV, respectively, are each within an alpha helix. Previous studies showed that a *Plasmodium* rhomboid protease cleaved TRAP proteins as a mechanism of gliding motility [40]. In *B. bovis*, rhomboid transcripts are differentially expressed between *Babesia* blood stages and kinetes. In blood stages, two rhomboid genes, BBOV_II005940 and BBOV_II005930, were upregulated, whereas two different rhomboid genes, BBOV_II006070 and BBOV_II006100, were upregulated in kinetes [18]. Therefore, we hypothesize that BBOV_II005940 and BBOV_II005930 cleave TRAP2, 3, or 4 for merozoite gliding and BBOV_II006070 and BBOV_II006100 cleave TRAP1 for kinete gliding in the tick hemolymph.

## 5. Conclusions

In conclusion, this study is the first to identify the stage-specific expression profile of *B. bovis* TRAP1, 2, 3, and 4 in blood stages, in vitro induced sexual stages, and kinetes. *Babesia bovis* TRAP2, 3, and 4 surface proteins may be important for parasite development in the mammalian host and tick midgut lumen, whereas *B. bovis* TRAP1 may play a key role in the infection of tick ovary cells. Further studies, including gene editing and antibody inhibition approaches, may elucidate the involvement of *B. bovis* TRAP proteins in parasite motility and invasion within the vertebrate and invertebrate hosts as seen with closely related Apicomplexan parasites that infect human beings. The results herein will contribute to the design of effective strategies to control infection of vertebrate and invertebrate hosts.

## Figures and Tables

**Figure 1 microorganisms-10-02173-f001:**
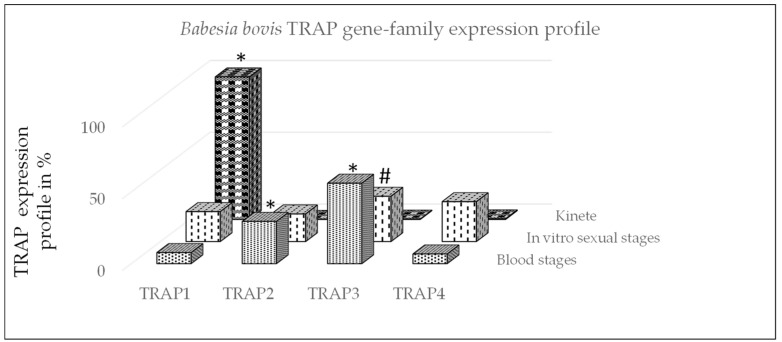
*Babesia bovis* gene-family expression profile using qPCR. The profile of TRAP genes was calculated for each stage by dividing each TRAP gene expression value by the total of TRAP transcripts and multiplied by 100 for a percentage and then summing the percentages within life stage per gene. *: significant within gene-family differences between stages (*p* < 0.05) and #: non-significant difference between blood and sexual stages (*p* = 0.06).

**Figure 2 microorganisms-10-02173-f002:**
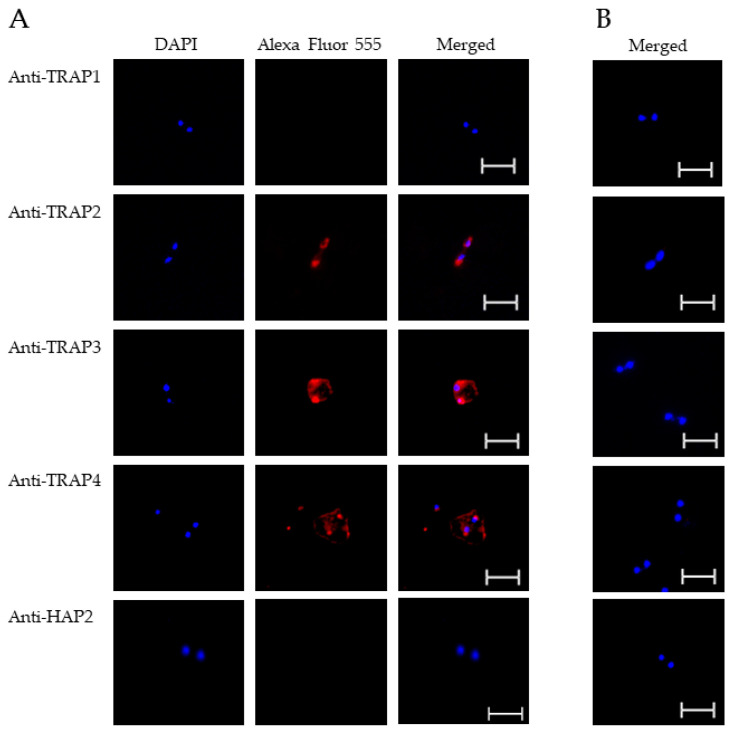
Evaluation of TRAP protein expression by *B. bovis* blood stages using a fixed immunofluorescence assay incubated with anti-TRAP 1, 2, 3, 4 or HAP2. Panels are (**A**) immune and (**B**) pre-immune rabbit serum. Blue indicates DAPI stained DNA and red indicates anti-rabbit IgG Alexa Fluor 555 reactivity. Scale bar: 5 μm.

**Figure 3 microorganisms-10-02173-f003:**
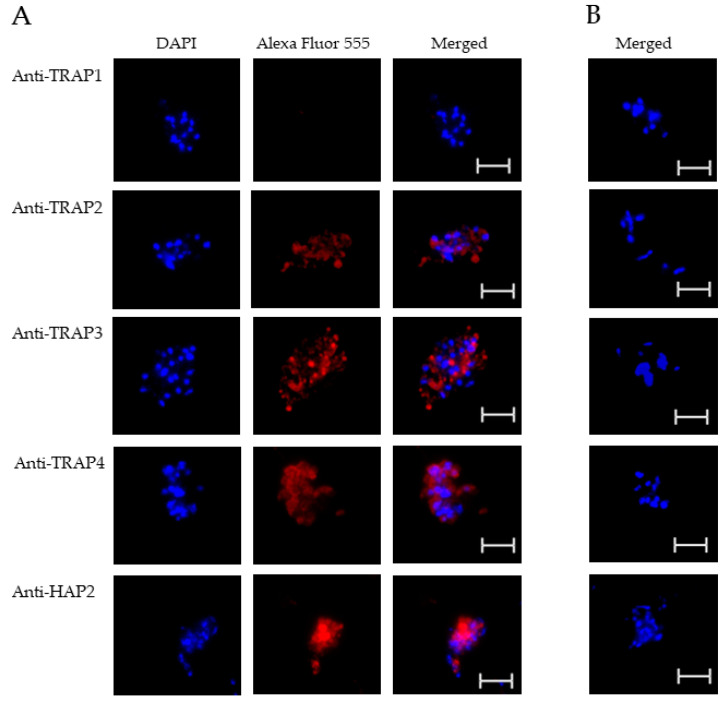
Evaluation of TRAP protein expression by *B. bovis* sexual stages using a fixed immunofluorescence assay incubated with anti-TRAP 1, 2, 3, 4, or HAP2. Panels are (**A**) immune and (**B**) pre-immune rabbit serum. Blue indicates DAPI stained DNA and red indicates anti-rabbit IgG Alexa Fluor 555 reactivity. Scale bar: 5 μm.

**Figure 4 microorganisms-10-02173-f004:**
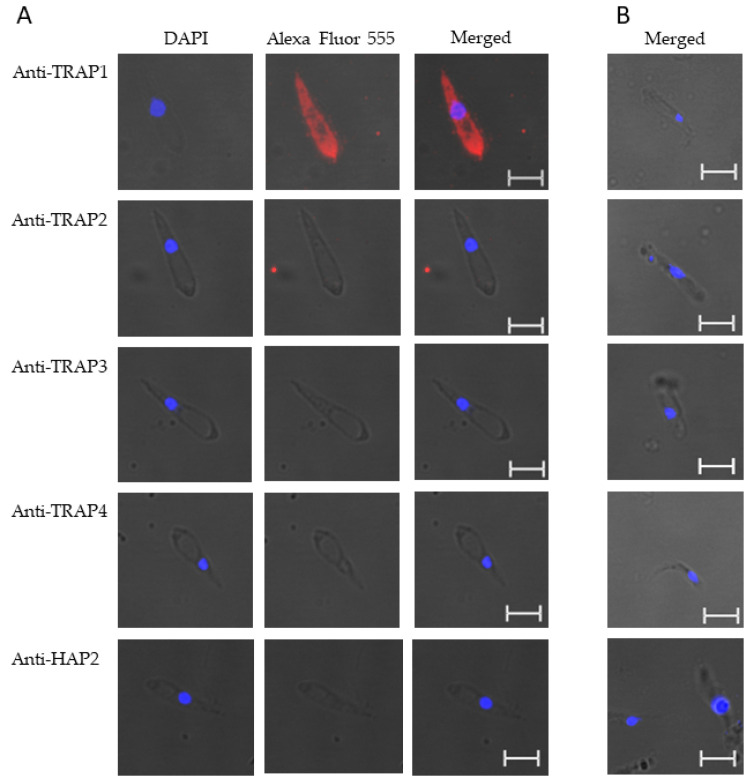
*Babesia bovis* kinetes expressed TRAP1 protein as determined by a fixed immunofluorescence assay incubated with anti-TRAP1, 2, 3, 4, or HAP2. Panels are (**A**) immune and (**B**) pre-immune rabbit serum. Blue indicates DAPI stained DNA and red indicates anti-rabbit IgG Alexa Fluor 555 reactivity. Scale bar: 5 μm.

**Figure 5 microorganisms-10-02173-f005:**
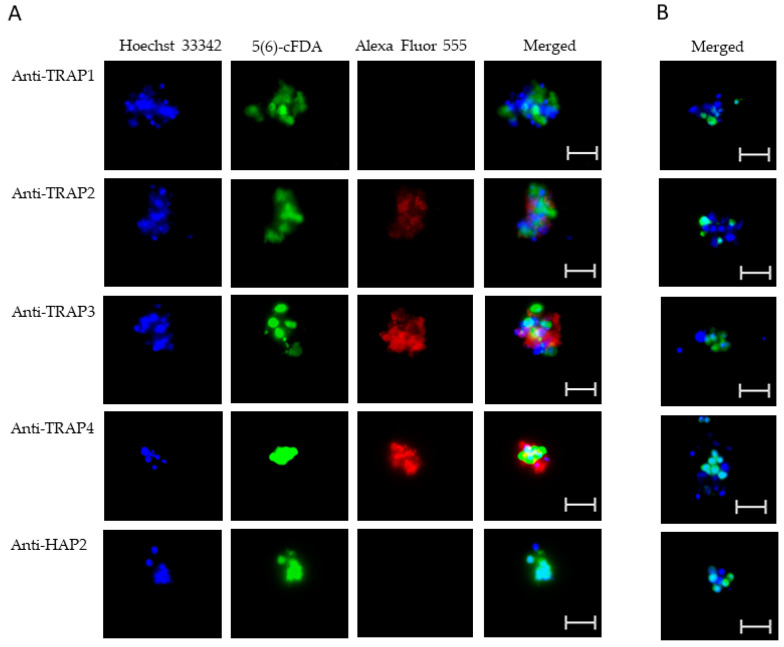
Evaluation of surface expression of TRAP proteins by *B. bovis* blood stages using a live immunofluorescence assay incubated with anti-TRAP 1, 2, 3, 4, or HAP2. Panels are (**A**) immune and (**B**) pre-immune rabbit serum. Blue indicates Hoechst 33342 stained DNA, green indicates live, intact exoerythrocytic parasites retaining 5(6)-cFDA, and red indicates anti-rabbit IgG Alexa Fluor 555 reactivity. Scale bar: 5 μm.

**Figure 6 microorganisms-10-02173-f006:**
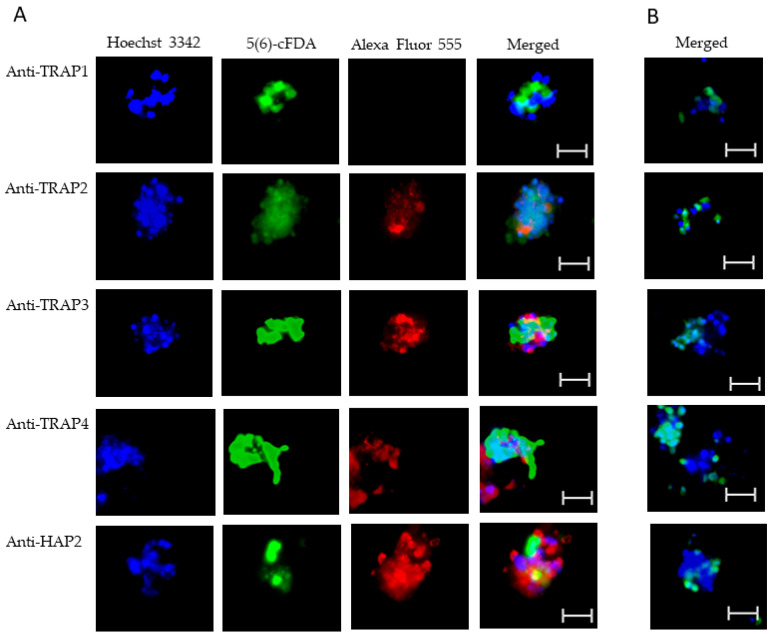
Evaluation of surface expression of TRAP proteins by *B. bovis* sexual stages using a live immunofluorescence assay incubated with anti-TRAP 1, 2, 3, 4, or HAP2. Panels are (**A**) immune and (**B**) pre-immune rabbit serum. Blue indicates Hoechst 33342 stained DNA, green indicates live, intact exoerythrocytic parasites retaining 5(6)-cFDA, and red indicates anti-rabbit IgG Alexa Fluor 555 reactivity. Scale bar: 5 μm.

**Figure 7 microorganisms-10-02173-f007:**
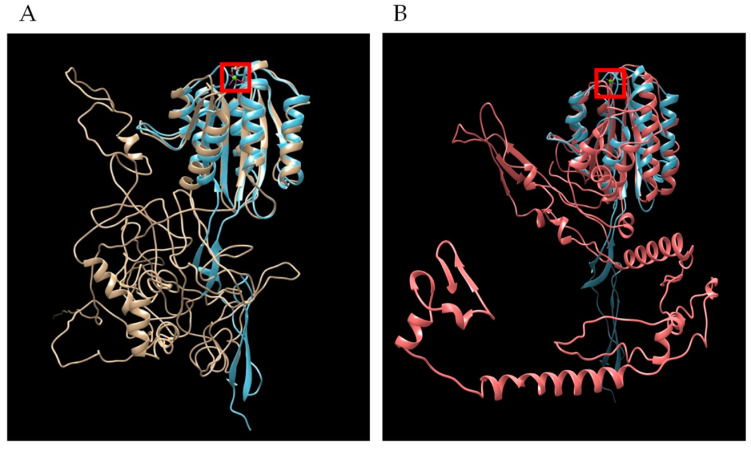
Overlap of secondary structures shared in tertiary models of *Plasmodium* TRAP with *B. bovis* TRAP. (**A**) *Plasmodium vivax* TRAP (4hqlA) is blue and *B. bovis* TRAP1 is gold and (**B**) *Plasmodium vivax* TRAP (4hqlA) is blue and *B. bovis* TRAP4 is red. Image generated by Chimera MatchMaker. Red square is Mg^2+^ ion bound by *P. vivax* TRAP.

**Table 1 microorganisms-10-02173-t001:** Quantitative PCR primers used to determine the number of transcripts expressed by *B. bovis* blood stages, in vitro induced sexual stages, and kinetes.

Gene ID	Protein Name	Sequence	TM
BBOV_II002650	TRAP1	F-5′ACC ACT TAC TCA ACT CCA ACT3′ R-5′GTA CCG GCC ATC CAA TCA A3′	55 °C 55 °C
BBOV_II002890	TRAP2	F-5′GTC ATG AGT ATT CCC AGC CTT C3′ R-5′TCA CTT CCT TCC GAT GCT TTC3′	55 °C 55 °C
BBOV_II002630	TRAP3	F-5′AAC CTA CCC AAA CCG GAA AT3′ R-5′AGT CGT TGT TAC TTG TCT CCT C3′	55.5 °C 55.5 °C
BBOV_II002870	TRAP4	F-5′TTT GGA TAC CGC TGT GCT AC 3′ R-5′CGG CTA GTC AAG GTC GTT AAA3′	55.5 °C 55.5 °C

**Table 2 microorganisms-10-02173-t002:** TRAP peptides used for rabbit immunization to generate specific antibodies against *B. bovis* TRAP proteins.

	Peptides		
TRAP1	ADKGVGSPKGKQC (28–40)	ESEDYEGEKQNDESNARSTSNTTK (571–594)	KKNKTPNETESGDYTGADESAE (616–638)
TRAP2	DRELSKVKVESEWYKPK (393–409)	ESERDSPVKNESTISSEIPK (899–910)	RGYIRLNRATREQSDIDENT (981–1000)
TRAP3	EASDKSAGAPSEDKSAESTSATE (752–774)	TNEHVETPAGTVESTESTSEEPTPVA (782–807)	KPEIETPSHEVAPTVDEHQN (944–963)
TRAP4	EHESTSLSRGPRPTEDQISQLPK (42–64)	ESSYRSRRLQSVEKHNEQQTGSQET (360–384)	NSGTHHPPHHRKGANGSGKK (462–481)

**Table 3 microorganisms-10-02173-t003:** TRAP family transcript expression levels in *B. bovis* blood, sexual, and kinetes stages.

	*Babesia bovis*		
	Blood Stages	In Vitro Sexual Stages	Kinetes
TRAP1	10^4.16 (4.3) b^	10^4.32 (3.6) b,c^	10^5.74 (4.8) b^
TRAP2	10^4.74 (4.3) c^	10^4.27 (3.6) b^	10^1.83 (4.8) a^
TRAP3	10^5.02 (4.3) a^	10^4.52 (3.6) a^	10^3.09 (4.8) a^
TRAP4	10^4.10 (4.3) b^	10^4.43 (3.6) a,c^	10^3.35 (4.8) a^

Pair-wise comparison demonstrated significant differences between gene expression within individual *B. bovis* stages. Different letters within Babesia stage represent significant differences between transcription levels of TRAP genes (*p* < 0.05).

**Table 4 microorganisms-10-02173-t004:** Conservation of residues of significance in *Plasmodium* sp. TRAP and *B. bovis* TRAP1 orthologs, with comparison to other *B. bovis* TRAP paralogs.

	Cysteine Residues Involved in MIDAS Domain Formation								Residues Involved in Mg^2+^ Binding				
*P. falcip* TRAP	C43	C205	C212	C235	C253	C257	C281	C286	D54	S56	S58	T131	D162
*P. vivax* TRAP	C39	C201	C208	C231	C249	C254	C277	C282	D50	S52	S54	T127	D158
*B. bovis* TRAP1	C40	C200	C206	C229	C248	C252	C283	C297	D51	S53	S55	T128	D158
*B. bovis* TRAP2	−	+	+	+	+	+	+	+	−	−	−	−	−
*B. bovis* TRAP3	−	+	+	+	+	+	+	+	−	+	−	−	−
*B. bovis* TRAP4	−	+	+	+	−	−	+	+	+	+	+	+	−

Residue name (C = Cysteine; D = Aspartic acid, S = Serine; T = Threonine) and location relative to individual TRAP polypeptide sequence. (+) indicates residue conserved in *B. bovis* TRAP paralog; (−) indicates residue not conserved in *B. bovis* TRAP paralog.

## Data Availability

The data presented in this study are available in Thrombospondin-Related Anonymous Protein (TRAP) family expression by *Babesia bovis* life stages within the mammalian host and tick vector.

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
