# Peer review of "Thrombospondin-Related Anonymous Protein (TRAP) Family Expression by Babesia bovis Life Stages within the Mammalian Host and Tick Vector"

_microorganisms, 2022, doi:10.3390/microorganisms10112173_

Round 1
Reviewer 1 Report
Well done study. All results are clearly described and properly discussed. Congratulation.
Just one small correction:
Lines 55-56: Plasmodium in italics
Author Response
We italicized as requested.
Reviewer 2 Report
In this study, the stage specific expression profile of Babesia bovis protein TRAPs in the blood phase, in vitro induction phase and in agonists was determined by immunofluorescence and quantitative PCR methods. Through this study, the authors believed that members of the TRAP family participated in the movement and invasion of parasites. I think this study is of some significance, but it seems far fetched to prove that the protein participates in the movement and invasion of parasites through the change of its expression during the infection stage. Therefore, I believe that if the authors want to prove that TRAP family proteins are of great significance to the movement and invasion of Babesia, they need other auxiliary proof experiments. The discussion part needs to further clarify the author's views, and the discussion part unrelated to this study can be removed.
Figure 1. It is unnecessary to use a stereoscopic map for this picture.
Table 1. It can be written into the method instead of using a table. Table 2 should use a three line table. Table 3 also needs to use a three line table like Table 2. In addition, there are two Table 3 in the text.
The negative sign in -80°C is incorrectly written.
Some spaces need attention. For example, "5 μm” not “5μm”
Author Response
To the Editor of the Microorganisms, on behalf of all the authors of our study entitled, " Thrombospondin-Related Anonymous Protein (TRAP) family expression by Babesia bovis life stages within the mammalian host and tick vector", we very much appreciate the reviewers’ comments. We addressed point by point all their concerns and modified the manuscript accordingly.
Comments:
In this study, the stage specific expression profile of Babesia bovis protein TRAPs in the blood phase, in vitro induction phase and in agonists was determined by immunofluorescence and quantitative PCR methods. Through this study, the authors believed that members of the TRAP family participated in the movement and invasion of parasites. I think this study is of some significance, but it seems far fetched to prove that the protein participates in the movement and invasion of parasites through the change of its expression during the infection stage. Therefore, I believe that if the authors want to prove that TRAP family proteins are of great significance to the movement and invasion of Babesia, they need other auxiliary proof experiments. The discussion part needs to further clarify the author's views, and the discussion part unrelated to this study can be removed.
Response: We agree with the reviewer. This manuscript did not test if TRAP proteins participate in gliding and invasion. We modified the discussion and conclusion sections for clarification.
Line 368-370: However, it is unknown if B. bovis TRAP proteins play important roles in parasite gliding movement or invasion of vertebrate and invertebrate host cells.
Line 442-451: In conclusion, this study is the first to identify the stage-specific expression profile of B. bovis TRAP1, 2, 3, and 4 in blood stages, in vitro induced sexual stages, and kinetes. Babesia bovis TRAP2, 3, and 4 surface proteins may be important for parasite development in the mammalian host and tick midgut lumen, whereas B. bovis TRAP1 may play a key role in the infection of tick ovary cells. Further studies, including gene editing and anti-body inhibition approaches, may elucidate the involvement of B. bovis TRAP proteins in parasite motility and invasion within the vertebrate and invertebrate hosts as seen with closely related Apicomplexan parasites that infect humans. The results herein will con-tribute to the design of effective strategies to control infection of vertebrate and invertebrate hosts.
Figure 1. It is unnecessary to use a stereoscopic map for this picture.
Response: We would like to keep the graphic as it is, because in our opinion it clearly shows differences between and within Babesia stages and individual genes.
Table 1. It can be written into the method instead of using a table. Table 2 should use a three line table. Table 3 also needs to use a three line table like Table 2. In addition, there are two Table 3 in the text.
Response: We would like to keep Table 1. It makes it easy and clear for the readers to see the primers used in this manuscript. All the tables were modified as recommended. We also edited Table numbers.
The negative sign in -80°C is incorrectly written.
Response: We have made changes as recommended throughout the manuscript.
Some spaces need attention. For example, "5 μm” not “5μm”
Response: We modified as recommended.
Reviewer 3 Report
The study is well designed, method is well described and results are well presented
I recommend it for publishing as it is
Author Response
Thank you.
Round 2
Reviewer 2 Report
The "-" at -80°C remains unchanged. I think the other parts have been properly modified.